# Website Investigation of Pet Weight Management-Related Information and Services Offered by Ontario Veterinary Practices

**DOI:** 10.3390/vetsci10120674

**Published:** 2023-11-27

**Authors:** Shawna Morrow, Kehan Zhang, Sarah K. Abood, Adronie Verbrugghe

**Affiliations:** Department of Clinical Studies, Ontario Veterinary College, University of Guelph, Guelph, ON N1G 2W1, Canada; smorrow@uoguelph.ca (S.M.); kehan@uoguelph.ca (K.Z.); sabood@uoguelph.ca (S.K.A.)

**Keywords:** online resources, canine nutrition, feline nutrition, veterinary marketing, veterinary professionals, pet obesity, weight loss

## Abstract

**Simple Summary:**

In addition to veterinary advice, pet owners rely on the internet for information on their pet’s health. Effectively managing a pet’s weight is often an underestimated component of wellness for pets, and there is an opportunity for veterinary practices to utilize an online platform to educate pet owners on the importance of weight management. The primary objective of this study was to describe the type of canine and feline weight management services, products, and information advertised or displayed on the websites of 50 veterinary practices in Ontario. An additional objective was to explore whether the size, company status, and location of the practice influences what is advertised or displayed. The preliminary results suggested that veterinary practices do not prioritize advertising weight management services, products, or educational material online for the public, and this was especially true for smaller practices with fewer veterinarians and veterinary technicians employed. Independently owned veterinary practices also seemed to advertise weight management products less than corporate practices. The exploratory findings of this study highlight the need for veterinary teams to provide educational and up-to-date online resources on weight management for pet owners. This, in turn, can provide trusted educational and accessible information for pet owners and contribute to client loyalty.

**Abstract:**

Pet owners rely on information and advice from their veterinary practice to effectively manage their pet’s weight. This study investigated weight management information and services displayed on practice websites in Ontario, Canada. Information collected from the websites of 50 randomly selected small and mixed-animal practices included practice and staff demographics and the type of weight management services, products, and information advertised or displayed. The most frequently advertised weight management service and product were nutritional counselling (34%) and therapeutic diets (25%), respectively. Current bodyweight measurement was advertised on just over half of the websites (54%), while physical therapy counselling was the least-advertised service (16%). Further statistical analyses were performed in an exploratory fashion to determine areas for future research. Binary logistic regression analyses were used to investigate the association between practice demographics and the type of weight management information advertised online. A maximum of two predictor variables were included in each regression model. Exploratory analyses indicated that when controlling for the number of veterinarians in each practice, having a higher number of veterinary technicians was associated with increased odds of a practice website advertising current bodyweight measurement by 80.1% (odds ratio (OR) = 1.80, *p* = 0.05). Additionally, when controlling the number of veterinary technicians, having a higher number of veterinarians was associated with increased odds of a practice website advertising sales of therapeutic diets by 119.0% (OR = 2.19, *p* = 0.04). When using corporate practices as reference, independently owned practices had decreased odds of advertising sales of treats and weight management accessories on their practice websites by 78.7% (OR = 0.21, *p* = 0.03). These preliminary results suggest that advertising weight management information is not prioritized on veterinary practice websites in Ontario, especially those with lower staff numbers. The findings of this study raise awareness on the current state of weight management promotion for pets on veterinary practice websites and highlight ways to improve upon a practice’s online presence.

## 1. Introduction

Weight management in dogs and cats, an essential component of companion animal medicine that is often undervalued or overlooked, refers to the process of adapting long-term lifestyle changes to maintain a pet’s healthy weight. This could involve weight loss and/or muscle and weight gain, while considering individualized factors such as breed, age, gender, and activity level of the pet [1,2]. An effective weight management plan includes both nutritional and physical therapy counselling to ensure appropriate caloric intake, diet selection, physical activity levels, and physical therapy where appropriate to ensure body fat loss and/or restore the use of muscles, bones, and the nervous system [3,4]. It also includes ways to modify the behaviors of both the owner and their pet to overcome barriers preventing them from reaching their goal [3]. Individualized weight management programs that provide consistent and healthy rates of weight loss or gain have been known to improve the quality of life for the animal by increasing lifespan and reducing the risk of or combatting already existing diseases or disorders, malnutrition, and certain types of cancers [3,4,5,6,7,8].

Despite knowledge that weight management is a vital component of wellness for companion animals, it remains a challenge for many pet owners to maintain a healthy weight in their pets. Canine and feline obesity is the most common nutritional disorder within veterinary practice and its prevalence has reached epidemic levels [4,9,10]. It is estimated that between 20 and 60% of adult cats and dogs in developed countries are overweight or obese [6,9,10,11,12,13,14,15,16]. The prevalence of underweight cats and dogs is less reported in the literature. However, recent studies have shown the prevalence of underweight adult dogs and cats in developed countries to be 4.2–11 and 5.3–10%, respectively [12,13,14,15,16,17]. Considering these high percentages, primarily among overweight cats and dogs, it is likely pet owners are unaware of the negative impact that an over-conditioned state can have on their pet’s health. For example, multiple studies have shown that pet owners continue to incorrectly believe their overweight or obese pet to be at an acceptable weight [18,19,20,21].

One way to combat this misunderstood aspect of healthcare in pets is to prioritize the promotion of weight management services and information in a method that is accessible and understandable for pet owners. Two recent studies were conducted using pet owner focus groups to understand pet owners’ expectations when it comes to information exchange with their veterinarian [22,23]. Many pet owners mentioned visiting the internet for additional information on their pet’s health following their veterinary appointment. They also expressed the desire for supplementary resources from their veterinarian, whether that be provided through the clinic or other reputable internet sources [22,23]. A recent online survey targeting pet owners in the United Kingdom found that the most common source for pet health information was the internet (449/571; 78.6%), followed by their veterinarian (441/571; 77.2%), yet veterinarians were considered the most trustworthy source of information [24]. An older study conducted by Hofmeister et al. (2008) showed that clients of veterinary practices consider the internet to be the third most popular source of information regarding pet health, following general practitioners and veterinary specialists [25]. Similar research from the United States has shown that 72.7% of pet owners surveyed (*n* = 1223/1683) considered the internet to be a supplemental source of information in addition to traditional vet care [26]. 

It is apparent that veterinary practice websites should provide informative and easy-to-understand resources and information about pet weight management, as well as the services and products they currently offer to assist in the weight management journey. Comprehensive and up-to-date websites can allow pet owners to gain confidence in their ability to manage their pet’s weight and to understand what assistance can be provided by their veterinary health care team. Accessible information can result in pet owners becoming more comfortable asking questions, ultimately leading to a sense of responsibility and self-motivation to improve their pet’s condition [24,26,27]. It can also help to establish veterinary clinics as trusted experts in pet health and wellness, further strengthening the relationship with pet owners and promoting client loyalty [22,23,24,25,26,27]. Finally, information regarding veterinary health care team members and their qualifications or interests in the field of pet weight management could be vital for pet owners when choosing the right veterinary practice for potential weight management consultations. A better understanding of what is advertised or displayed on veterinary practice websites is necessary to overcome current barriers pet owners face when understanding the importance of and pursuing weight management for their pets.

The authors hypothesized that online marketing of canine and feline weight management services is not a priority for companion animal veterinary practices in Ontario. Therefore, the primary objective of this study was to investigate the type of weight management services and information that are advertised or displayed on the websites of veterinary practices in Ontario and the frequency with which they are promoted. An exploratory analysis was also performed to investigate whether any veterinary practice and staff demographics influence the type of weight management services and information advertised or displayed on clinic websites. This exploratory analysis was meant to serve as an initial look into the influence of demographic factors on the online advertising abilities and/or priorities of veterinary practices and to suggest which factors might be important to consider for future research. 

## 2. Materials and Methods

### 2.1. Data Collection

A total of 50 small and mixed-animal Ontario practice websites were investigated. The practices were selected from a list of 783 practices that have referred at least one patient (dog or cat) to the Ontario Veterinary College’s Health Science Centre (OVC HSC) and were within a 75 km radius from the Ontario Veterinary College in Guelph, Ontario. Practices were randomly selected using a computer-generated randomization list via www.randomization.com. The radius was chosen to ensure 50 practices could be randomly selected. The veterinary practices were not aware of the investigation at any time. The websites were initially investigated from January to March 2022 and re-investigated by a third party from November to December 2022. The websites were re-investigated to include a more in-depth search of weight management information for the purpose of this study. The only data used for this study was obtained from the most recent website investigation, and comparisons were not made between each time point.

Information collected was separated into three categories: practice demographics, staff demographics, and pet weight management services and information displayed. All survey questions can be found in Appendix A. For this investigation, the only information used was that made available through websites, except for certain demographic information. 

Veterinary practice demographic information included the name, address, location (urban, suburban, or rural), type of practice (general, emergency and specialty, and mixed), species served, and the practice’s company status (independently owned or belonging to a corporation). The name of the practice was entered into an internet search engine to access the practice’s website. If the address was not listed on the website, the information was taken from the internet search engine. The location (urban, suburban, rural) was derived from the practice’s address. Definitions used to define a practice as urban, suburban, or rural can be found in Appendix A. If it was not explicitly stated on the website that the practice belonged to a corporation or was independently owned, lists of practices associated with all known veterinary corporations in Ontario were searched. If the practice was not listed, it was deemed an independently owned practice. 

Staff demographic information included the total number of staff and number of veterinarians and veterinary technicians listed on the website. The number of veterinarians and veterinary technicians with nutrition and physical therapy credentials was also recorded, as well as whether any staff members had additional training in veterinary nutrition, as noted in the staff biographies on the website. Credentials were accepted when they consisted of an ongoing or completed board certification, graduate degree, or training program included in the American Association of Veterinary State Boards’ Registry of Approved Continuing Education Programs focused on nutrition or physical therapy. Accepted nutrition and physical therapy credentials for veterinarians and veterinary technicians can be found in Appendix A. Additional training for staff members in veterinary nutrition included those considered nutrition advisors and those with nutrition certificates from pet food companies or other online nutrition certifications.

The pet weight management services, products, and information displayed on the practice websites included the advertising of a weight management service, nutritional counselling, physical therapy counselling, current bodyweight measurement, and sales of veterinary therapeutic diets, treats, food puzzles, or measurement tools and weight management accessories such as leashes and toys, and whether they displayed educational material on weight management. Links to webstores as well as information directly displayed on the website were used to investigate whether the practice advertised sales of therapeutic diets, treats, food puzzles, or measurement tools and weight management accessories. 

Note that for this study, the term “canine physical therapy” is used, which is interchangeable with the term “canine rehabilitation”. Canine physical therapy encompassed both conventional physical therapy and complementary and alternative therapies. Physical therapy consists of conventional evidence-based treatments that help the patient to restore the use of muscles, bones, and the nervous system. Treatment modalities include physical activity, manual massage, passive range of motion, walking, hydrotherapy, joint mobilization, and heat and cold therapy [28]. Complementary and alternative therapies consist of a range of therapies that are not part of current standard veterinary medical practice but can be used in addition to conventional treatments. Treatment modalities include laser therapy, therapeutic ultrasound, acupuncture, and transcutaneous or neuromuscular electrical stimulation [29]. Additionally, for this study, nutritional counselling encompassed information on body composition assessments, calculating ideal bodyweight, collecting a diet history, and recommendations on diet selection, feeding amount, feeding frequency, feeding management, daily treat allowance, and supplementation [5]. 

### 2.2. Data Analysis 

Descriptive statistics, including frequency (*n*) and percentage (%) data, were performed in Microsoft Excel version 16.71 for all veterinary practice demographics (location, company status, practice type, and type of species served), staff demographics (total number of staff, number of veterinarians and veterinary technicians, number of veterinarians and veterinary technicians with nutrition and physical therapy credentials, and whether any staff members had additional nutrition training), and pet weight management information and services displayed on the websites (whether the website advertised a weight management service, nutrition and physical therapy counselling, current bodyweight measurement, and sales of therapeutic diets, treats, food puzzles, or measurement tools and accessories and displayed educational material). 

All other statistical analyses were performed in R Studio, version 4.2.2 (31 October 2022). Statistical models were selected based on the nature of the dependent variables of interest, which in this case were all binary. To determine the overfitting parameter, the number of independent (predictor) variables entered into the models was chosen based on the smallest category of the dependent variables (present/absent). The smallest category of the following dependent variables of interest allowed for a maximum of two predictor variables per model: advertising a weight management service, current bodyweight measurement, and sales of therapeutic diets, treats, and accessories and displaying educational material. The smallest category of the following dependent variables of interest allowed for a maximum of one predictor variable per model: advertising nutritional counselling. The smallest category of the following dependent variables did not allow for any predictor variables to be included in the models and were therefore not run: advertising physical therapy counselling and the sale of food puzzles and measurement tools. 

Binary logistic regression models were designed to determine the likelihood that veterinary practice and staff demographics influenced the odds of a practice advertising pet weight management information and services. Three models were developed for the following dependent variables, measured as a binary present/absent outcome: advertising a weight management service, current bodyweight measurement, and sales of therapeutic diets, treats, and accessories. Independent variables considered in the first model included the number of veterinarians and veterinary technicians (continuous variables) listed on each practice website. Veterinarians and veterinary technicians were included in the same model as there is likely a correlation between the number of veterinarians and veterinary technicians employed within each practice. By including both staff members in the model, the influence of the number of veterinarians employed on certain outcome variables could be examined while controlling for the number of veterinary technicians, and vice versa. A linear relationship between the two numerical independent variables (number of veterinarians and veterinary technicians) and the logit transformation of each dependent variable was determined upon visual inspection through scatter plots. The independent variable considered in the second model was the location of the practice (3 categorical levels: urban, suburban, or rural). Including categorical data in a binary regression model reduces the number of predictor variables allowed by one (N-1 degrees of freedom); therefore, location was the only independent variable permitted. The independent variable considered in the third model was the company status of the practice (dichotomous variable: belonging to a corporation or independently owned). Two models were developed for the dependent variable, displaying educational material (measured as a binary present/absent outcome). The independent variable considered in the first model was location, and company status was considered in the second model. The numbers of veterinarians and veterinary technicians were not included as independent variables in a model as the linearity assumption was not met upon visual inspection. Four models were developed for the following dependent variable, measured as a binary present/absent outcome: advertising nutritional counselling. Each model consisted of one independent variable due to the overfitting parameter, and the following independent variables were used: number of veterinarians and number of veterinary technicians listed on the websites, location, and company status of the practices. For all analyses that considered location as the independent variable, two regression models were run. The first model used urban practices as the reference variable, which allowed the comparison between urban and suburban, and urban and rural practices. The second model used suburban practices as the reference variable, which allowed the final comparison between suburban and rural practices. For all analyses that considered company status as the independent variable, belonging to a corporation was used as the reference variable. In total, 28 binary logistic regressions were run.

Certain veterinary practice (practice type, type of species served) and staff (number of veterinarians and veterinary technicians with nutrition or physical therapy credentials, whether any staff members had additional nutrition training) demographic information could not be included as independent variables in the models due to the lack of variation in the data. Most practices provided general care for small animals (46/50) with only four providing emergency services and one providing care for small and large animals. There was also not enough variation in the type of species served, as all practices serviced cats and dogs, and only 15 serviced pocket pets and 4 serviced pocket pets and exotic species, along with cats and dogs. Finally, most practice websites did not report any veterinarians or veterinary technicians having nutrition and/or physical therapy credentials, or any staff members with additional nutrition related training. 

To understand the proportion of the variance in the dependent variables that can be explained by the predictor variables (number of veterinarians and veterinary technicians, location, and company status) in the models, pseudo R^2^ values were reported. Pseudo R^2^ values were interpreted through the CoxSnell, McFadden, and Nagelkerke models, and a range of variance was reported. The statistical software used for analysis did not allow for the interpretation of R^2^ values, so the pseudo R^2^ values reported should be interpreted with caution. Statistical significance was set at *p* ≤ 0.05; trends were recognized if *p* was >0.05 but <0.1. All tables and figures were created in GraphPad Prism 9.5.1.

Due to the large number of tests run, there is potential for an inflated type 1 error. Considering the exploratory nature of these analyses, it was chosen not to correct for multiple comparisons. The purpose of the regression analyses was to have an initial look into the potential influence of veterinary practice and staff demographics on the advertising of weight management information online. Therefore, the results should mainly be interpreted as insight for future, larger studies, and caution should be taken when examining the unadjusted *p*-values. Of the 28 binary regression analyses run, 5 were identified as having relevance for future research and are discussed below. 

## 3. Results

### 3.1. Demographic Information

To compare demographic information collected in this study to the greater population, the number of active companion animal veterinary practices and veterinarians in Ontario was collected from the College of Veterinarians of Ontario (Table 1) [30]. The number of active registered veterinary technicians in Ontario was collected from the Ontario Association of Veterinary Technicians, and this number could not be filtered by species (Table 1) [31].

#### 3.1.1. Veterinary Practice Demographics

All 50 veterinary practices investigated served small animals, with 45 (90%) general practices, and 5 (10%) emergency or specialty clinics. There were no practices investigated that served both small and large animals. Many served only cats and dogs (34/50; 68%), while 15 (30%) and 4 (8%) served pocket pets and exotics along with cats and dogs, respectively. Only one (2%) practice served only cats. Information regarding the location and company status of the practices is summarized in Table 2. 

#### 3.1.2. Staff Demographics

Of the veterinarians and veterinary technicians listed on the practice websites, only two veterinarians reported holding any nutrition credentials in their biography. Additionally, very few veterinarians and veterinary technicians reported holding credentials in the field of veterinary physical therapy. Moreover, additional training in veterinary nutrition was not mentioned in most of the staff members’ biographies (Table 2).

### 3.2. Weight Management Services, Products, and Information Displayed

More than half of the veterinary practice websites advertised nutritional counselling (66%; Figure 1A), and just over half advertised having a weight management service and current bodyweight measurement (54%; Figure 1A). Educational material related to weight management was displayed on just under half of the websites (46%; Figure 1A). Physical therapy counselling was the least-advertised weight management service (*n* = 8/50; 16%; Figure 1A). Regarding the sale of weight management products, therapeutic diets were advertised most frequently (50%; Figure 1B). The same veterinary practices that advertised the sale of treats also advertised the sale of weight management accessories, and these were advertised on just under half of the practices (42%; Figure 1B). The sale of food puzzles and/or measurement tools was not advertised on any of the practice websites.

### 3.3. Influence of Veterinarians and Veterinary Technicians on Displaying Weight Management Services, Products, and Information on Websites

On average, when controlling for the number of veterinarians in each practice, having a higher number of veterinary technicians was associated with increased odds of a practice website advertising current bodyweight measurement by 80.1% (Table 3). When controlling for the number of veterinary technicians in each practice, the number of veterinarians did not have an influence on the advertising of current bodyweight measurement on practice websites. Regarding the advertising of current bodyweight measurement, pseudo R^2^ values revealed that 16.8–27.6% (Appendix A) of the variance in the data could be explained by the combined number of veterinarians and veterinary technicians working in the practice.

Additionally, on average, when controlling for the number of veterinarians in each practice, having a higher number of veterinary technicians tended to increase the odds of a practice website advertising a weight management service in general by 63.5% (OR = 1.63, *p* = 0.08). When controlling for the number of veterinary technicians in each practice, the number of veterinarians did not have an influence on the advertising of a weight management service on practice websites. Regarding the advertising of a weight management service, 13.9–23.3% (Appendix A) of the data can be explained by the combined number of veterinarians and veterinary technicians working in the practice.

On average, when controlling for the number of veterinary technicians in each practice, having a higher number of veterinarians was associated with increased odds of a practice website advertising the sale of therapeutic diets by 119.0% (Table 3). When controlling for the number of veterinarians in each practice, the number of veterinary technicians did not have an influence on the advertising of selling therapeutic diets on practice websites. Pseudo R^2^ values revealed 17.4–27.8% (Appendix A) of the variance in the data could be explained by the combined number of veterinarians and veterinary technicians working in the practice. Binary logistic regressions did not reveal any influence of the number of veterinarians and veterinary technicians working in the practice on whether the practice websites advertised the sale of treats and weight management accessories. 

### 3.4. Influence of Veterinary Practice Company Status on Displaying Weight Management Services, Products, and Information on Websites

In terms of weight management services, binary logistic regressions did not reveal any influence on whether practice websites advertised a weight management service, nutritional counselling, and current bodyweight measurement between practices belonging to a corporation and those that were independently owned. 

Using corporate veterinary practices as the reference, independently owned practices had decreased odds of advertising the sale of treats and weight management accessories on the practice websites, on average, by 78.7% (Table 4). Pseudo R^2^ values revealed that 7.9–13.7% (Appendix A) of the variance in the data could be explained by the company status of the veterinary practices. Binary logistic regressions revealed that being independently owned or belonging to a corporation did not influence whether a veterinary practice advertises the sale of therapeutic diets or the displaying of educational material on weight management on its website. 

### 3.5. Influence of Veterinary Practice Location on Displaying Weight Management Services, Products, and Information on Websites

Binary logistic regressions did not reveal any influence of the location (urban vs. suburban; urban vs. rural; suburban vs. rural) of a veterinary practice on whether the practice website advertised a weight management service, nutritional counselling, current bodyweight measurement, and sales of therapeutic diets, treats, and accessories. 

It was also investigated whether the location of a veterinary practice influenced whether the practice website displayed educational material in written, video, or blog format. When using urban practices as the reference, residing in a rural area tended to decrease the odds of a veterinary practice website displaying educational material by 86.1% (OR = 0.56, *p* = 0.08). There was no difference in displaying educational material on practice websites between urban and suburban practices. When using suburban practices as the reference, there was no difference in displaying educational material on practice websites between suburban and rural practices. Pseudo R^2^ values revealed that 6.3–11.1% (Appendix A) of the variance in the data could be explained by the location of the veterinary practices in both models. 

## 4. Discussion

The results of this study help gain insight into the current state of accessible and reliable information that is available online for pet owners to aid in their knowledge and decision making when it comes to their pet’s weight management. On the veterinary practice websites evaluated in the present study, the most-advertised weight management service was nutritional counselling. Similarly, the most-advertised products were veterinary therapeutic diets, followed by treats and weight management accessories such as leashes and toys. This finding is supported by previous research investigating the types of pet obesity information that are present or absent online [32]. Making changes to a pet’s diet to aid in weight management and the recommendation of a weight loss diet were mentioned most often [32]. Another qualitative study found that the most frequent pet health information pet owners search online for is medical concerns and diet/nutritional information [23]. The weight management guidelines put forth by the American Animal Hospital Association (AAHA) primarily focus on the nutritional component of weight loss or gain and recommend that nutritional assessments be performed regularly by the veterinary team. Additionally, the World Small Animal Veterinary Association (WSAVA) along with the AAHA have classified nutrition as the fifth vital sign following temperature, pulse, respiration, and pain [4,33,34]. Companion animal nutrition has been well studied, with research leading to detailed dietary and feeding guidelines [4,5]. The strength of these recommendations from the AAHA and WSAVA, along with the current state of research, should lead veterinary teams to feel most confident in pursuing the nutritional aspect of weight management with their patients. Considering this perceived importance of nutrition in the health and well-being of pets, it may explain its prominence on veterinary practice websites when compared to other weight management services and products.

Measuring a patient’s current bodyweight is an important first step in determining a weight management plan in pets [1,33], yet it was only advertised by half of the websites as a service. This aligns with the study mentioned above conducted by Chen et al. in 2020, who found that less than half of the online sources investigated mentioned measuring a pet’s weight, with even fewer mentioning how to properly weigh a pet or describing a body condition score chart [32]. It is known that body condition scoring (BCS) and bodyweight measurements are two of the most common methods used by veterinarians to assess body composition [1]. These valuable tools can be used by both veterinary staff and pet owners to determine a pet’s ideal bodyweight. Determining a pet’s ideal bodyweight could help motivate pet owners to effectively manage their pet’s weight with a tangible goal in mind [1,4]. However, multiple studies have shown that pet owners continue to inaccurately determine their pet’s body condition score using the BCS chart [18,19,20,21]. A questionnaire sent to pet owners revealed that about half of the respondents were able to correctly estimate their dog’s bodyweight using electronic scales [21]. Veterinary practices may want to avoid the possibility of providing information online that has the potential to be misused, and as a result, they avoid advertising the service altogether. Nevertheless, pet owners play an important role in the monitoring of their pet’s health and wellbeing [35]. Providing online tools to help pet owners complete an accurate body composition assessment can contribute to the overall success of weight management in pets. Advertising the current bodyweight measurement can also allow for veterinary practices to raise awareness of its importance in the weight management journey. To further encourage the proper use of weighing scales and body composition assessment tools, detailed instructions could accompany all online resources to provide a step-by-step guide for pet owners.

Physical therapy appeared to be the least-prioritized service in terms of advertisement. In the present study, only a small number of veterinary practice websites advertised any type of physical therapy, including physical activity. In a similar study evaluating online pet obesity information, a small number of online sources advertised physical therapy, and making dietary changes was addressed significantly more for weight loss than increasing levels of physical activity [32]. In the present study, weight management accessories that aid in promoting physical activity, such as leashes and toys, were advertised on less than half of the practice websites. To date, there are limited studies investigating the benefits of incorporating physical therapy into a weight management plan for pets [36,37,38]. Increasing physical activity is often recommended by veterinarians in conjunction with dietary changes [39], but research regarding the effectiveness of such a program is scarce. In contrast to nutritional recommendations, there is a lack of evidence to help determine an ideal physical activity program for cats and dogs [4]. Determining the caloric expenditure for different types of physical activity is relatively unexplored, aside from walking in dogs [4,40]. A survey sent to veterinary colleges within the United States and Canada also revealed that less than half offered a dedicated course in integrative veterinary medicine, a combination of complementary and alternative therapies with conventional care. Of those who offered courses in this area, all but one was offered as an elective [41]. The lack of published research and guidelines on the use of physical therapy for weight management, along with the minimal education for veterinarians, could help to explain the low advertisement of this service on practice websites. Regardless, practice websites could benefit from having a section dedicated to physical therapy and/or physical activity. If available, veterinarians with board certifications focusing on nutrition and physical therapy could assist with content creation. Knowledge dissemination related to weight loss protocol in relation to physical therapy is closely related to expertise from these fields. 

Aside from services and products, providing pet owners with educational information and resources on pet weight management is a trusted and accessible way to promote proper pet weight management care. A recent study using pet owner and veterinarian focus groups found that pet owners expressed the desire for information to be explained in multiple ways, and many deemed using a visual aid along with verbal explanation, as well as being directed to reputable internet sources, to be effective [22]. Veterinarians in all focus groups also felt that providing additional resources to pet owners is important [22]. Owners in another recent focus group study expressed the desire for veterinarians to provide trusted online resources to supplement the content learned during the appointment [23]. Unfortunately, in the present study, only about half of the veterinary practice websites displayed some type of educational material on weight management, whether that be in written or video format, or links to external sources. This aligns with previous research findings that few pet owners receive recommendations for online resources from their veterinarians, despite their willingness to use such recommendations [23,24,42,43]. Perhaps since both veterinarians and pet owners express the benefits in having informative and trustworthy online resources, additional efforts should be considered by all veterinary team members in creating educational online content. Examples of ways to achieve this include creating original content, subscribing to online software that provides educational material for veterinary practices, and providing links to trusted external sources. 

Exploratory analyses suggested that when controlling for the number of veterinarians working in the practice, the number of veterinary technicians influenced the advertising of current bodyweight measurement and a weight management service. Limited studies examining the role of the veterinary technician during weight management assessments exist. A recent qualitative study was conducted to understand the perceptions of veterinary professionals in relation to their practice’s weight management service. Many respondents believed that the qualified veterinary technicians within their practice provided nutrition- and weight-related information, most commonly following the veterinarians [44]. Another preliminary study targeting veterinary professionals, primarily veterinary technicians, noted that most respondents would initiate a discussion with the owner regarding their pet’s weight, regardless of weight status, and many indicated that they would provide recommendations on caloric intake, measuring food, and exercise [45]. It is important to note that results of these studies were not separated for each practice, and the number of veterinarians and veterinary technicians working at each practice was unknown [44,45]. Preliminary results from the present study may suggest that utilizing the expertise and skills of veterinary technicians can further assist in promoting and raising awareness on the topic of weight management. Future research should investigate the benefits of delegating the promotion and advertising of weight management care to trained technicians or other veterinary support staff.

Exploratory analyses also indicated that when controlling for the number of veterinary technicians in practice, the number of veterinarians influenced the advertising of selling therapeutic diets. A recent survey given to small-animal veterinary health care team members noted that veterinarians were the most common source of nutrition-related information for pet owners, followed by veterinary nurses/technicians. There was also a significant relationship between the frequency with which veterinarians performed nutritional assessments and the establishment of a normal dietary regime, calculation of energy requirements, and formulation of nutritional plans [45]. Research also suggests that veterinarians remain the leading source of information when it comes to pet owners purchasing pet food [44,45,46,47]. If veterinarians and veterinary technicians are the two main sources of nutritional and dietary-related services/education, veterinary practices that employ a high number of these staff members may have increased advocacy for the advertisement of nutritional counselling and products such as therapeutic diets and treats online. Additional studies should be performed on a larger scale to determine the impact of the size of the practice, specifically in terms of the number of veterinarians and veterinary technicians employed, on the advertising and promotion of nutritional and weight management products in general. 

A final exploratory finding worth noting from this study was the influence of a veterinary practice’s company status on the advertisement of products. Practices that are independently owned had decreased odds of advertising treats and weight management accessories compared to those belonging to a corporation. This is especially interesting as less than a third of the veterinary practices belonged to a corporation, and the remainder were independently owned. Incorporation has risen in popularity in recent years. Belonging to a corporation may increase opportunities available for veterinary practices, in that staff members can focus on veterinary medicine rather than the management of a business and subsequent marketing. Additional resources might be available for advancement in the layout of corporate practice websites, including pre-existing templates that ensure available products are present online, as well as links to any online veterinary stores [48]. This could also include shared educational tools and handouts that are available for all practices within the corporation to help advertise products and services, and to help educate pet owners on all topics related to weight management. To build upon these preliminary results, studies could focus on the comparison of website design between independently owned practices and those belonging to a corporation. These studies could also determine whether a relationship exists between advertising of services and products and profit of sales. 

There were a few limitations to this study, the first being that information presented on the websites may not accurately reflect what is offered in person in veterinary practices. It was unknown whether the websites were complete and up to date regarding all aspects of the practice. Statistical analyses were performed with an exploratory purpose to determine areas for future research, and there is the potential for an inflated type 1 error as the results have not been corrected for multiple comparisons. Both the sample size and geographical radius of veterinary practices chosen were small, which limited the number of inferential statistics that could be run, to avoid overfitting the model. The small number of predictor variables permitted in the models also did not allow for the inclusion of co-variates or confounders, which might have influenced the results. Possible co-variates could include the education level/credentials of veterinarians and veterinary technicians in the field, the type of practice (emergency, specialty, etc.), and the species served. Possible confounders include the availability of services and/or products, knowledge of website design, and the economic status and level of establishment of the practice. Although the practices were uninformed of this investigation and were chosen at random, there is a possibility that they became aware of the study and modified their websites accordingly. A final limitation is that veterinary practices were chosen from a list of referring practices to the Ontario Veterinary College, which may have resulted in some practices within the 75 km radius who had never referred a patient before to the college to be missed. Considering the small sample size, extrapolation of findings from this study to the greater population of Ontario should be made with caution. However, regardless of sample size, this study should act as a call for more research and action from veterinary practices and staff members to improve upon the sources of information provided to pet owners regarding weight management. A broader website search should be conducted within a wider geographical range and without the prerequisite of being a referring practice to avoid sampling bias. Further research could also include comparing information found on the websites of veterinary practices to what is truly offered in-clinic to evaluate online accessibility for pet owners seeking veterinary care or advice. The findings of this study highlight the need for veterinary practices to improve upon their weight management promotion. Providing content online can increase awareness about pet obesity and help owners more effectively manage their pet’s weight from home.

## 5. Conclusions

With the growing use of the internet for pet health information, veterinary practices are presented with the opportunity to utilize an online platform to raise awareness on the importance of weight management in pets. However, based on the results, it seems that veterinary practices in Ontario are not prioritizing the advertisement of weight management resources frequently, and there remains room for improvement. Services and products related to nutrition were advertised most frequently, with little priority given to the promotion of physical therapy and physical activity in relation to weight care. Educational resources on weight management were also not provided to pet owners on many websites. Exploratory analyses indicated that future research should consider the influence of a practice’s size, location, and company status on the frequency with which they promote weight management services, products, and educational material online. The findings of this study raise awareness on the current state of weight management promotion for pets on veterinary practice websites and highlight ways to improve upon a practice’s online presence. This can ensure pet owners have access to up-to-date information and trusted resources they can rely on.

## Figures and Tables

**Figure 1 vetsci-10-00674-f001:**
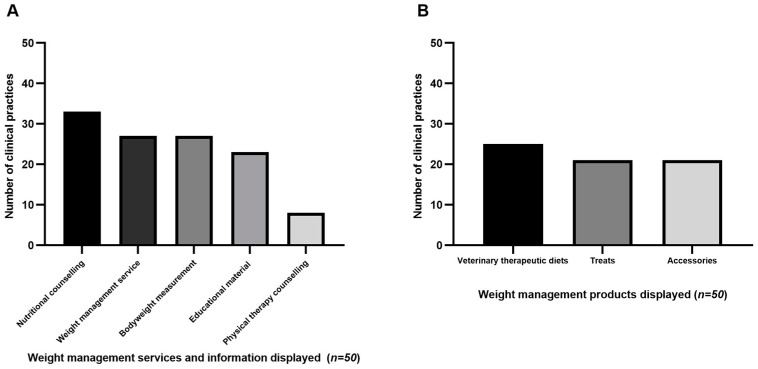
Weight management services and information displayed (**A**) and weight management products displayed for sale (**B**) on websites of 50 veterinary practices in Ontario.

**Table 1 vetsci-10-00674-t001:** Demographic information relating to the number of veterinary practices, veterinarians, and veterinary technicians in this study and in Ontario.

	# of Veterinary Practices	# of Veterinarians	# of Veterinary Technicians
In this study	50	162	166
In Ontario	1257	5383	4496

**Table 2 vetsci-10-00674-t002:** Veterinary practice and staff demographic information displayed on websites of 50 veterinary practices in Ontario. Staff demographic data are based on the total number of veterinarians (*n* = 162) and veterinary technicians (*n* = 166) mentioned on all veterinary practice websites.

Category	Variable	%	*n*
Veterinary practice demographics	Located in urban area	66	33
	Located in suburban area	20	10
	Located in rural area	14	7
	Belonging to a corporation	26	13
	Independently owned	74	37
	Have staff with nutrition-related training	4	2
Staff demographics (*n* = 162 veterinarians and 166 veterinary technicians)	Veterinarians with nutrition credentials	4	2
	Veterinarians with physical therapy credentials	0.62	1
	Veterinary technicians with nutrition credentials	4.3	7
	Veterinary technicians with physical therapy credentials	0	0

**Table 3 vetsci-10-00674-t003:** Binary logistic regression models exploring the association between (model 1) whether current bodyweight measurement is advertised on practice websites (yes/no outcome) and the number of veterinarians and veterinary technicians working in the practice and (model 2) whether the sale of therapeutic diets is advertised on practice websites (yes/no outcome) and the number of veterinarians and veterinary technicians working in the practice. Bolded values indicate significance (*p* < 0.05).

Variable	Odds Ratio	95% CI	Std. Error	*p*-Value
Model 1 Current bodyweight measurement				
# of veterinarians (controlling for # of veterinary technicians	0.73	0.36–1.37	0.33	0.349
# of veterinary technicians (controlling for # of veterinarians)	1.80	1.09–3.66	0.30	**0.049**
Model 2 Selling therapeutic diets				
# of veterinarians (controlling for # of veterinary technicians)	2.19	1.10–5.15	0.39	**0.042**
# of veterinary technicians (controlling for # of veterinarians)	0.82	0.48–1.37	0.26	0.448

**Table 4 vetsci-10-00674-t004:** Binary logistic regression model exploring the association between whether the sale of treats and weight management accessories is advertised on practice websites (yes/no outcome) and the company status of the veterinary practice. Bolded values indicate significance (*p* < 0.05).

Variable	Odds Ratio	95% CI	Std. Error	*p*-Value
Independently owned practices (reference = practices belonging to a corporation)	0.21	0.05–0.79	0.70	**0.026**

## Data Availability

The data presented in this study are available in the University of Guelph Research Data Repository at https://doi.org/10.5683/SP3/QIDLGF.

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
