# Peer review of "Website Investigation of Pet Weight Management-Related Information and Services Offered by Ontario Veterinary Practices"

_vetsci, 2023, doi:10.3390/vetsci10120674_

Round 1
Reviewer 1 Report (Previous Reviewer 2)
Comments and Suggestions for Authors
Dear Editors,
the authors have taken into consideration the majority of my comments and suggestions. I therefore suggest that you accept the publication of this article in its present form.
Yours faithfully,
Author Response
Comment: Dear Editors,
the authors have taken into consideration the majority of my comments and suggestions. I therefore suggest that you accept the publication of this article in its present form.
Response: We would like to extend our thanks and appreciation to this reviewer and are glad to hear they are satisfied with the way we addressed their comments and suggestions.
Reviewer 2 Report (New Reviewer)
Comments and Suggestions for Authors
Thank you for the opportunity to review this manuscript on pet weight management.
The authors have done an online research to study available data to pet owners on management of their pets. Overall the study is well thought and the results interesting. The authors state that the aim of the study is to guide future studies, which is acceptable in view of the limited information currently.
I only have a few points of discussion for the authors:
1. Consider having your odd ratios summarized in a table and include the confidence interval. This more useful than the P value by itself.
2. Line 133: There is mention of two investigations of the practices at 10 month interval. However, it is not clear if there was a difference between both time points or how the results changed between the two time points. THis should be expanded/explained.
3. Table 3: the numbers are not lined up with the variables and difficult to read.
4. Table 4 is missing.
Author Response
Comment: Thank you for the opportunity to review this manuscript on pet weight management.
The authors have done an online research to study available data to pet owners on management of their pets. Overall the study is well thought and the results interesting. The authors state that the aim of the study is to guide future studies, which is acceptable in view of the limited information currently.
I only have a few points of discussion for the authors:
Response: We would like to extend our thanks for this review to the reviewer. We have responded to their suggestions below.
Comment: 1. Consider having your odd ratios summarized in a table and include the confidence interval. This more useful than the P value by itself.
Response: Thank you for this suggestion. We also believe that including odds ratios and confidence intervals in a table will be valuable for readers. We have added this information for the exploratory results that were deemed significant, and can be found on lines 377-382, and 398-401. To account for the additional information added, we have removed the summary of the odds ratio, and p-value within the text portion of the results to ensure we are not providing the same information twice. These changes can be found on lines 351, 368, and 392.
Comment: 2. Line 133: There is mention of two investigations of the practices at 10 month interval. However, it is not clear if there was a difference between both time points or how the results changed between the two time points. This should be expanded/explained.
Response: Thank you for bringing our attention to this, and we are happy to provide further clarification. We have added the following sentence into the methods section (lines 139-142), and hope it is satisfactory: “Websites were re-investigated to include a more in-depth search of weight management information for the purpose of this study. The only data used for this study was obtained from the most recent website investigation, and comparisons were not made between each time point.”
Comment: 3. Table 3: the numbers are not lined up with the variables and difficult to read.
Response: We appreciate you catching this formatting error. As per another reviewer’s comments, we have converted the information presented in Table 3 to a figure. Therefore, there is no longer any formatting issue as Table 3 has been removed. The new figure can be found on lines 340-345. We also included frequency and percentage information for all categories in the text of section 3.2, to account for the removal of this information from the table (lines 328,329,332,333,335,336).
Comment: 4. Table 4 is missing.
Response: We apologize for this inconvenience. As per a previous review, it was suggested to remove the table as results were double presented in the table and the text. We had therefore removed the table but forgot to remove the reference to the table in the text. We do however believe the table to be valuable for readers for visual purpose and to get a better understanding of the odds ratios and confidence intervals associated with the p-values and have re-included the table. To account for this, as mentioned above, results presented in the table was removed from the text portion of the results. Also, please note that the inclusion of a figure altered the numbering of the tables. The previous Table 4 is now Table 3.
Reviewer 3 Report (New Reviewer)
Comments and Suggestions for Authors
Dear Authors,
Review can be found below.
Authors acknowledge study limitations – that conclusions drawn from websites and not employees can lead to false conclusions derived from limited information collected.
The questionnaire that would be sent to those practices would be a good solution, and a comparison could be made between advertised and real expertise offered.
The article is well written and statistical analysis is sufficient.
In the discussion, authors are using outdated citation forms describing the cited articles' findings. Therefore, the discussion can be rewritten so that it is more concise and clear. For example, sentence lines 481-485 could be rephrased more concisely.
The statement in lines 506-509 is too strong and does not have a scientific basis and should be revised.
Compared to veterinarians who usually follow a more biomedical approach to practice, veterinary technicians prefer a more patient-centered approach, which may lead to further discussions and educational opportunities for pet owners on weight management.
Again in lines 515 – 518 a claim that is too strong as weight loss protocol and physical therapy is a process that needs to be supervised by veterinarian.
Results from the present study may suggest that employing a larger number of veterinary technicians could present an opportunity to utilize their knowledge of weight management in pets to a larger degree, hence increased the advertisement of the services. Future research should investigate the benefits of delegating the promotion of weight management care to trained technicians or other veterinary support staff.
An integral part of the ACVIM nutrition specialist – board-certified, ACVIM nutrition residents and American College of Veterinary Sports Medicine and Rehabilitation specialists (and European colleges) should be briefly discussed – as knowledge dissemination related to weight loss protocol in relation to physical therapy is closely related to expertise from those fields.
Nutrition credentials from Table S1 Chinese food/herb therapy (Chi Institute) cannot be used in the same category as ACVIM nutrition specialists (or ECVCN). Residents should be included in that category.
Chinese food/herb therapy (Chi Institute) credentials should be briefly and critically discussed in the manuscript.
For Veterinary Technicians – the same applies.
For American College of Veterinary Sports Medicine and Rehabilitation specialists, it should be placed in different categories as ACVAA (American College of Veterinary Anesthesia and Analgesia), CVPP (Certified Veterinary Pain Practitioner), CVA (Certified Veterinary Acupuncture, Chi Institute), ACVA (The American Veterinary Chiropractic Association, CoAC (College of Animal Chiropractors).
Author Response
Comment: Authors acknowledge study limitations – that conclusions drawn from websites and not employees can lead to false conclusions derived from limited information collected.
The questionnaire that would be sent to those practices would be a good solution, and a comparison could be made between advertised and real expertise offered.
The article is well written and statistical analysis is sufficient.
Response: We would like to extend our thanks for this review to the reviewer. We have responded to their suggestions below.
Comment: In the discussion, authors are using outdated citation forms describing the cited articles' findings. Therefore, the discussion can be rewritten so that it is more concise and clear. For example, sentence lines 481-485 could be rephrased more concisely.
Response: Thank you for helping to ensure we use the most up-to-date citation forms and remain concise. We have altered lines 481-485 to reflect the changes suggested and can be found on lines 502-506. We have also changed a few other citation formats to remain as concise as possible, and these changes can be found on lines 431-433, 457-459, 476-479, and 472-475. We hope these changes are satisifactory.
Comment: The statement in lines 506-509 is too strong and does not have a scientific basis and should be revised.
Compared to veterinarians who usually follow a more biomedical approach to practice, veterinary technicians prefer a more patient-centered approach, which may lead to further discussions and educational opportunities for pet owners on weight management.
Response: We appreciate this comment and agree that the statement was too strong based on the evidence at hand. We added more relevant information to this section, lines 522-531, and believe it to better compare to our results and reflect on the point we are trying to make. We therefore made the decision to remove the statement you were concerned with: “Compared to veterinarians who usually follow a more biomedical approach to practice, veterinary technicians prefer a more patient-centered approach, which may lead to further discussions and educational opportunities for pet owners on weight management”.
Comment: Again in lines 515 – 518 a claim that is too strong as weight loss protocol and physical therapy is a process that needs to be supervised by veterinarian.
Results from the present study may suggest that employing a larger number of veterinary technicians could present an opportunity to utilize their knowledge of weight management in pets to a larger degree, hence increased the advertisement of the services. Future research should investigate the benefits of delegating the promotion of weight management care to trained technicians or other veterinary support staff.
Response: We appreciate your concern regarding the statement above. However, our intended message was that future research could investigate the benefits in using veterinary technicians or other support staff to promote the advertising of weight management care, not actually provide the care themselves. We can see how this may not have been clear in the statement written, and we have changed the sentence to better reflect what we meant to say. It now reads (lines 531-535): “Preliminary results from the present study may suggest that utilizing the knowledge of veterinary technicians can further assist in promoting and raising awareness on the topic of weight management. Future research should investigate the benefits of delegating the promotion and advertising of weight management care to trained technicians or other veterinary support staff.”
Comment: An integral part of the ACVIM nutrition specialist – board-certified, ACVIM nutrition residents and American College of Veterinary Sports Medicine and Rehabilitation specialists (and European colleges) should be briefly discussed – as knowledge dissemination related to weight loss protocol in relation to physical therapy is closely related to expertise from those fields.
Response:
We agree that board-certified veterinarians in the field of nutrition and physical therapy are valuable in disseminating knowledge related to physical therapy for weight loss. However, we do believe that investigating how they can contribute to improved knowledge dissemination is outside the scope of this paper, as most veterinarians did not include having any credentials in their biographies. We did include a statement on lines 492-495, explaining how we can better utilize the knowledge of these individuals when it comes to promoting physical therapy. It reads: “Veterinarians with board-certifications focusing on nutrition and physical therapy could assist with content creation. Knowledge dissemination related to weight loss protocol in relation to physical therapy is closely related to expertise from these fields.” We hope this accurately portrays what was suggested.
Comment: Nutrition credentials from Table S1 Chinese food/herb therapy (Chi Institute) cannot be used in the same category as ACVIM nutrition specialists (or ECVCN). Residents should be included in that category.
Chinese food/herb therapy (Chi Institute) credentials should be briefly and critically discussed in the manuscript.
For Veterinary Technicians – the same applies.
For American College of Veterinary Sports Medicine and Rehabilitation specialists, it should be placed in different categories as ACVAA (American College of Veterinary Anesthesia and Analgesia), CVPP (Certified Veterinary Pain Practitioner), CVA (Certified Veterinary Acupuncture, Chi Institute), ACVA (The American Veterinary Chiropractic Association, CoAC (College of Animal Chiropractors).
Response: Thank you for bringing this to our attention, and we understand the need to separate credentials into different categories. In Table S1, we have created a new category “Additional nutrition credentials”, for both veterinarians and veterinary technicians. We separated board-certified credentials from all others. We have included the following statement on lines 166-169: “Credentials were accepted when they consisted of an ongoing or completed board-certification, graduate degree or training program included in the American Association of Veterinary State Boards’ Registry of Approved Continuing Education Programs focused on nutrition or physical therapy.” We did not specifically discuss the Chi Institute in detail but are hoping that the statement on lines 166-169 fulfills this request, as certifications from the Chi Institute are included in the registry of approved continuing education programs.
We have also separated the American College of Veterinary Sports Medicine and Rehabilitation specialists’ credential from the other credentials. ACVAA (American College of Veterinary Anesthesia and Analgesia), CVPP (Certified Veterinary Pain Practitioner), CVA (Certified Veterinary Acupuncture, Chi Institute), ACVA (The American Veterinary Chiropractic Association, CoAC (College of Animal Chiropractors), are now in a separate category entitled “Additional physical therapy credentials”. Residents, candidates, and ongoing training for all nutrition and physical therapy credentials was also included. Our investigation did include staff members who were residents, candidates, or had ongoing training, but we did not find any staff members with this specification in their biographies and therefore did not include it in our definitions. The data repository has been updated and these changes should now be visible in Table S1, and the new DOI link is as follows: https://doi.org/10.5683/SP3/QIDLGF, as been updated in the manuscript on lines 636-637.
Reviewer 4 Report (New Reviewer)
Comments and Suggestions for Authors
The paper is well written, with no recommendations for any changes.
Author Response
Comment: The paper is well written, with no recommendations for any changes.
Response: We would like to extend our thanks and appreciation to this reviewer and are glad to hear they are satisfied with the manuscript.
Reviewer 5 Report (New Reviewer)
Comments and Suggestions for Authors
The primary objective of this study was to determine the type of weight management services and information that is advertised or displayed on websites of veterinary practices in Ontario, and the frequency with which they are promoted.
Major issues
Were the veterinary practices aware of the investigation? This should be mentioned clearly in M&M. If they were not, was there any possibility that they found out somehow about the investigation and modified their websites accordingly? A passage should be included in the Discussion.
The analysis is OK, although the authors could have performed multivariate analysis to improve the conclusion of the study. However, it is OK, as it is.
Table 4 is missing, which was a hindrance to correct evaluation.
Discussion is OK, but please add a passage on how the findings will influence future clinical services on weight management.
Minor issues
Sub-section 3.1. should be reduced in length, as that is not the main scope of the study.
Can I suggest to add some graphs in order to visualise the findings? This will be a great help for future readers.
There are one or two recent relevant references that the authors may find useful to add to complement their line of thoughts.
Please do add no ideas in the concluding section; in there, just recapitulate the findings and their interpretation.
Author Response
Comment: Major issues
Were the veterinary practices aware of the investigation? This should be mentioned clearly in M&M. If they were not, was there any possibility that they found out somehow about the investigation and modified their websites accordingly? A passage should be included in the Discussion.
Response: We would like to extend our thanks to the reviewer for their thoughtful review.
We appreciate you bringing our attention to this potential limitation. The veterinary practices were not aware of the investigation at any time, and we have addressed this in the methods on line 137. We also included a passage in the discussion reflecting this potential limitation on lines 586-589.
Comment: The analysis is OK, although the authors could have performed multivariate analysis to improve the conclusion of the study. However, it is OK, as it is.
Response: We appreciate your concern regarding our statistical analyses but appreciate your understanding of the methods we chose to use. To further your understanding, we have included the reasoning (provided in a previous review) as to why we could not include multivariate analysis below:
We met with one of our statistics experts to see if it would be possible to run the regressions with multiple variables as you suggested, and we explain below what we did and why we chose to not include all predictor variables into one model based on our definition of overfitting.
As explained by our statistician and from our own research, overfitting works differently for binary logistic regression, which is what we ran as our dependent variables are binary (yes/no option). For binary logistic regression, the 1 in 10 rule is applied to the dependent variables, rather than the independent. For each dependent variable, we looked at which variable, either the “yes” or the “no” category had the smallest count, and then divided that number by 10 to determine how many predictor variables we could have in our model. For example, when looking at whether the practice had a weight management service as a dependent variable, there were 27 practices that had the service (our “yes” category) and 23 that did not have the service (our “no” category). Therefore, the “no” category has the smallest count at 23, so we divided 23 by 10 to get 2.3. Rounding this down to 2, it told us that we can have a maximum of two predictor variables within the model to avoid overfitting. If we include more than 2 variables, our results will be so specific to our own dataset that it would be difficult to generalize to the larger population (Ontario/Canada). Based on the number of yes’s and no’s, we can also only include two predictor variables for the models involving whether the practice advertised measuring current bodyweight, diets, treats, accessories, and educational material. For nutritional counselling, physical therapy counselling, and food puzzles, our smallest category of yes’s and no’s was too small to allow for us to have more than one predictor variable, so we can’t run this with multiple independent variables in the model. We referenced this paper when determining how many Independent variables can fit in the model: Babyak, MA., What You See May Not Be What You Get: A Brief, Nontechnical Introduction to Overfitting in Regression-Type Models, Psychosomatic Medicine 66:411-421 (2004).
We agree that when determining the influence of vets and vet techs, they should be included in the same model to account for the fact they are from the same practice. In terms of location as one of our independent variables, it is a categorical variable with three options (urban, suburban, rural). Since it is categorical, we have to subtract the number of predictor variables allowed in the models by 1 (n-1 degrees of freedom). Since our models are all allowed two predictor variables, we had to subtract one, leaving us with only one predictor variable allowed when using location as an independent variable. We re-ran the models in the following way:
- DV: weight management service; IV: vets + vet techs
- DV: weight management service; IV: Location
- DV: weight management service; IV: company status
- This was repeated for all other dependent variables that allowed for two predictor variables (bodyweight, diets, treats, accessories)
Comment: Table 4 is missing, which was a hindrance to correct evaluation.
Response: We apologize for this inconvenience. As per a previous review, it was suggested to remove the table as results were double presented in the table and the text. We had therefore removed the table but forgot to remove the reference to the table in the text. We do, however, believe the table to be valuable for readers for visual purposes and to get a better understanding of the odds ratios and confidence intervals associated with the p-values and have re-included the table. To account for this, as mentioned above, results presented in the table were removed from the text portion of the results. Also, please note that the inclusion of a figure altered the numbering of the tables. The previous Table 4 is now Table 3.
Discussion is OK, but please add a passage on how the findings will influence future clinical services on weight management.
Response: Thank you for this suggestion. We have added the following sentences to the end of the discussion (lines 600-603) to highlight how the findings will influence future clinical services: “The findings of this study highlight the need for veterinary practices to improve upon their weight management promotion. Providing content online can increase awareness about pet obesity, and help owners more effectively manage their pet’s weight from home.”
Comment: Minor issues
Sub-section 3.1. should be reduced in length, as that is not the main scope of the study.
Response: We appreciate you finding ways to keep the manuscript as concise as possible, and we agree that sub-section 3.1 can be shortened. We have shortened the text in sections 3.1 (lines 297-302), 3.1.1 (lines 311-312), and 3.1.2 (315-319), and hope this is satisfactory.
Comment: Can I suggest to add some graphs in order to visualise the findings? This will be a great help for future readers.
Response: Thank you for providing a suggestion to help make this manuscript more appealing to future readers. We have removed Table 1 and replaced the information in figure format (lines 339-345). We thought this was the best information to provide in a figure format as it is the easiest to visualize. We also included frequency and percentage information for all categories in the text of section 3.2, to account for the removal of this information from the table (lines 328,329,332,333,335,336).
Comment: There are one or two recent relevant references that the authors may find useful to add to complement their line of thoughts.
Response: We would be happy to include all relevant references in the manuscript to compliment our line of thoughts. Upon searching the internet for recently published papers that align with our manuscript, we found one relevant reference that we wanted to include. This reference involved pet owner focus groups to understand pet owner’s expectations when exchanging information with veterinarians, and is discussed on lines 83-83, 431-433, and 503-505. We also added additional information related to pet owner and veterinarian perceptions of communication on lines 499-502.
If there are other relevant references you had in mind that we missed, we would be happy to discuss them and would appreciate if you provided us with the reference or doi to the paper.
Comment: Please do add no ideas in the concluding section; in there, just recapitulate the findings and their interpretation.
Response: Thank you for this suggestion. We have removed all ideas from the conclusion and added more information related to our findings and ways to interpret them. This can be found on lines 609-612.
Round 2
Reviewer 5 Report (New Reviewer)
Comments and Suggestions for Authors
The authors have addressed all issues thoroughly and have made extensive changes to improve the manuscript, which is now ready for acceptance.
I have no further comments.
Author Response
We would like to thank this reviewer for all of their suggestions to help improve upon our manuscript, and are glad we addressed their comments to their satisfaction.
This manuscript is a resubmission of an earlier submission. The following is a list of the peer review reports and author responses from that submission.
Round 1
Reviewer 1 Report
Comments and Suggestions for Authors
The work is well structured and the objective of the research is adequate and practical.
I like the manuscript and in my opinion it is easy to read and understand
My only objection is the small sample size.
I would like clear indications for clinical veterinarians to increase their effectiveness in this field.
Small improvements in these senses should be included in this manuscript.
Author Response
Comment: The work is well structured and the objective of the research is adequate and practical.
I like the manuscript and in my opinion it is easy to read and understand
Response: We would like to extend our thanks for this review to referee 1 and we have responded to their comments and questions below.
Comment: My only objection is the small sample size.
Response: We appreciate your concern when it comes to the sample size of the study. This research initially began as an undergraduate research project, and the small sample size was chosen to ensure the undergraduate student could complete the data collection, analysis, and writing of the paper by the end of the school year. As we are aware the sample size may not be representative of a more general population (i.e., Ontario/Canada), we have added the following statement into our limitations section of the discussion: “Considering the small sample size, extrapolation of findings from this study to the greater population of Ontario should be made with caution. However, regardless of sample size, this study should act as a call for more research, and action from veterinary practices and staff members to improve upon the sources of information provided to pet owners regarding pet weight management.”
Comment: I would like clear indications for clinical veterinarians to increase their effectiveness in this field.
Response: Thank you for this suggestion. We have added clear indications for clinical veterinarians to increase their effectiveness in this field throughout our discussion, and they can be found on lines 484-487, 509-514, 555-557,586-588, 609-623.
Reviewer 2 Report
Comments and Suggestions for Authors Dear authors,
I congratulate you on the quality of your work.
I have a few suggestions, which I think would add value to your article:
1- Add a map with the distribution of veterinary practices and veterinary practitioners concerned by your study.
2- Add information on the number of veterinary practitioners, technicians and practices in the country and in Ontario.
3- You can use statistical analyses such as multivariate analyses to better discuss the results of your study.
We wish you all the best.
Author Response
Comment: Dear authors,
I congratulate you on the quality of your work.
I have a few suggestions, which I think would add value to your article:
Response: We appreciate these kind words and would like to extend our thanks to referee 2. We have responded to their suggestions below.
Comment: 1- Add a map with the distribution of veterinary practices and veterinary practitioners concerned by your study.
2- Add information on the number of veterinary practitioners, technicians, and practices in the country and in Ontario.
Response: Thank you for these helpful suggestions. We interpreted these suggestions as providing information regarding the number of veterinary practices and veterinary practitioners (and veterinary technicians) within our general population in which we derived our sample size from. In this case, Ontario, and Canada. For Ontario, we found information from the College of Veterinarians of Ontario on the number of active companion animal hospitals, and the number of active veterinarians that serve cats and dogs. We also found the number of active registered veterinary technicians in Ontario from the Ontario Association of Veterinary Technicians, although we could not filter these for the type of species they serve. We included the number anyways. For Canada, we could not find information related to the number of companion animal veterinary practices and veterinarians. We only found information related to the total number of practices and veterinarians in Canada (small animal, large animal, exotic, etc.), so this information is not applicable to our study and did not include it in our manuscript. We also could not find any information related to the total number of veterinary technicians in Canada. We have added the information stated above in the results section, lines 278-284.We have also added this information in a table to compare the numbers to our own study. Table 1 compares our study to the Ontario population of veterinary practices, veterinarians, and veterinary technicians with the information available to us on the internet.
Comment: 3- You can use statistical analyses such as multivariate analyses to better discuss the results of your study.
Response: Thank you for bringing our attention to multivariate analysis. We have responded to this comment in depth above, when replying to the academic editor’s comments. We hope the information provided is sufficient, and our rationale for the inclusion of two vs. one predictor variables into the models is adequate.